https://doi.org/10.1038/s41467-018-07803-9　　**OPEN**

# The gut microbiome is required for full protection against acute arsenic toxicity in mouse models

Michael Coryell[1], Mark McAlpine[1], Nicholas V. Pinkham[1], Timothy R. McDermott[2] & Seth T. Walk[1]

Arsenic poisons an estimated 200 million people worldwide through contaminated food and drinking water. Confusingly, the gut microbiome has been suggested to both mitigate and exacerbate arsenic toxicity. Here, we show that the microbiome protects mice from arsenic-induced mortality. Both antibiotic-treated and germ-free mice excrete less arsenic in stool and accumulate more arsenic in organs compared to control mice. Mice lacking the primary arsenic detoxification enzyme (As3mt) are hypersensitive to arsenic after antibiotic treatment or when derived germ-free, compared to wild-type and/or conventional counterparts. Human microbiome (stool) transplants protect germ-free As3mt-KO mice from arsenic-induced mortality, but protection depends on microbiome stability and the presence of specific bacteria, including *Faecalibacterium*. Our results demonstrate that both a functional As3mt and specific microbiome members are required for protection against acute arsenic toxicity in mouse models. We anticipate that the gut microbiome will become an important explanatory factor of disease (arsenicosis) penetrance in humans, and a novel target for prevention and treatment strategies.

[1] Department of Microbiology & Immunology, 109 Lewis Hall, Montana State University, Bozeman, MT 59717, USA. [2] Department of Land Resources and Environmental Sciences, 637A Leon Johnson Hall, Montana State University, Bozeman, MT 59717, USA. Correspondence and requests for materials should be addressed to S.T.W. (email: seth.walk@montana.edu)

Arsenic is a toxic metalloid and human carcinogen, ranking first on the US Agency for Toxic Substances and Disease Registry and the US Environmental Protection Agency Priority List of Hazardous Substances for the past 20 years (since 1997)[1]. Arsenic toxicity varies widely depending on the chemical composition of arsenic-containing compounds (arsenicals). Trivalent species are more toxic than pentavalent species, although the toxicity of the latter can increase depending on whether and to the extent it becomes thiolated. The microbiomes of humans and mice were shown to metabolize arsenic when cultured in vitro[2-4]. Similarly, mice with different microbiome compositions were found to excrete different types of arsenicals following exposure to arsenic[5,6]. However, no study has provided direct in vivo evidence that the mammalian microbiome either exacerbates or protects the host from arsenic-related pathology or disease.

In a series of experiments utilizing antibiotic-treated, transgenic, germ-free (GF), and gnotobiotic mouse models, we illustrate the critical contribution of the gut microbiome in protecting the host from acute arsenic toxicity. We provide evidence that microbiome stability during arsenic exposure is a key determinant of survival to arsenic exposure; link important bacterial taxa from the human microbiome with protection; and show that the human gut commensal, *Faecalibacterium prausnitzii*, is sufficient for at least some protection against arsenic. The overall conclusion from this study is that the gut microbiome is a significant factor underlying survival to acute arsenic toxicity.

## Results

**Microbiome disruption affects arsenic excretion/bioaccumulation.** We initially defined the microbiome's role in arsenic detoxification in vivo by experimentally disrupting the bacterial community of laboratory-reared, wild-type (WT) mice with the cephalosporin antibiotic, cefoperazone (Cef). This drug decreases total bacterial load in mice by ~3 orders of magnitude and significantly alters taxonomic composition[7]. Cef was administered in drinking water for 2 days prior to and throughout a 14-day exposure to inorganic sodium arsenate (iAs$^V$), the most common arsenic species in human drinking water. Groups of mice were exposed to 25 and 100 ppm iAs$^V$ based on (i) an established dose-equivalent equation that adjusts for surface area differences between mice and humans[8]; (ii) the use of similar exposures in studies addressing arsenic cytotoxicity in mice[9-12]; and (iii) documented human drinking water exposures[13] (Supplementary Note 1 and Supplementary Table 1). In short, our approach was designed to evaluate arsenic toxicity in the context of high, acute exposure.

Significantly less arsenic was excreted in the stool of Cef-treated mice (87 ± 6% less in 25 ppm group and 93 ± 3% less in 100 ppm group) compared to Sham-treated counterparts (Fig. 1a). These results suggest that more arsenic was retained in Cef-treated mice, and evaluation of organs confirmed an overall arsenic accumulation ($p < 0.0001$, 25 ppm; $p = 0.0003$, 100 ppm; two-way ANOVA). The experimental design was not powered to determine whether specific organs were more likely to accumulate arsenic, but we did observe significant accumulation in lung ($p < 0.0001$) and liver ($p = 0.025$) after correcting for multiple comparisons (Fig. 1b). Together, these results support the hypothesis that the microbiome mitigates host exposure and uptake of ingested iAs$^V$ through fecal excretion.

**As3mt and microbiome are required for protection.** The above mice showed few signs of disease with only two exceptions (two Cef-treated mice in the 100 ppm group were found dead). This is consistent with at least two other studies of WT mice, where no mortality was observed during exposure to 25 and 100 ppm iAs$^V$ [10,14] or to the more toxic, trivalent arsenite (iAs$^{III}$) for at least 4 weeks[10]. In contrast to WT mice, arsenic (+3 oxidation state) methyltransferase-deficient mice (As3mt-KO) are dysfunctional in arsenic methylation[15], accumulate more intracellular inclusions in urothelial tissue during arsenic exposure[10], and have increased overall arsenic-induced cytotoxicity[9,12]. We hypothesized that As3mt-KO mice would be more sensitive to high doses of arsenic compared to WT mice, and that toxicity would increase following microbiome perturbation. iAs$^V$-induced mortality in As3mt-KO mice at 25 ppm iAs$^V$ exposure has not been reported, and as expected, no toxicity was observed in Sham-treated As3mt-KO mice exposed to 25 ppm iAs$^V$ for up to 22 days. However, the same level of exposure was lethal in half of Cef-treated As3mt-KO mice by day 12 (Fig. 2a). At 100 ppm iAs$^V$, mortality was observed in both Sham- and Cef-treated groups (Fig. 2b), but was more rapid in mice with a disrupted microbiome ($p = 0.003$, Mantel–Cox test).

To quantify survival in the absence of a microbiome, a GF line of As3mt-KO mice was derived and were exposed to increasing amounts of iAs$^V$. Dose-dependent mortality was observed in mice exposed to 10, 25, and 100 ppm iAs$^V$ (Fig. 2c), suggesting toxicity increased with increased exposure. For comparison, a GF line of WT mice was exposed to iAs$^V$ (25 and 100 ppm). Mortality was observed in only 2 of 22 WT mice exposed to 100 ppm iAs$^V$ out to 40 days (one each on days 19 and 38). Therefore, both As3mt and an intact gut microbiome are necessary for full protection against arsenic toxicity in this mouse model. Finally, we tested the hypothesis that Cef treatment somehow increased iAs$^V$ toxicity. The timing of antibiotic pre-treatment in conventional As3mt-KO mice was changed and the survival of GF As3mt-KO mice was quantified with and without Cef treatment (Supplementary Fig. 1). There was no evidence that Cef increased iAs$^V$ toxicity, and so iAs$^V$ toxicity was likely the sole determinant of mortality.

**Human stool transplantation provides full protection.** Bacteria in human stool possess genes encoding arsenic-active enzymes[16] that can biochemically transform arsenic in vitro[3,4,17], and genes encoding arsenic-active enzymes are present in human gut bacterial genome databases[16]. Beyond such direct enzyme–arsenic interactions, bacteria might also provide indirect benefits to the host during arsenic exposure by producing metabolites that otherwise enable excretion and tissue repair. However, the ability of the human gut microbiome to provide protection to a mammalian host has never been quantified. Given their hypersensitivity to arsenic, GF As3mt-KO mice were used to determine whether a human microbiome could provide protection from arsenic. Stool from a healthy human donor was transplanted into these mice prior to iAs$^V$ exposure. The human donor was not knowingly exposed to arsenic and drank water from a regulated municipal source in the US. The transplanted stool was established in GF recipient As3mt-KO mice (referred to as F0) for 10 days prior to arsenic exposure. In parallel, the same stool was transplanted into gravid, GF As3mt-KO dams, so that offspring (F1) could develop with this microbiome from birth. We hypothesized that F1 mice would have increased protection against arsenic due to early-life development in the presence of a human microbiome. In experiments where F1 mice were reared to approximately the same age at exposure as F0 mice, transplanted human stool provided full protection against 100 ppm iAs$^V$, regardless of the timing of microbiome acquisition (Fig. 3a) and the median survival of both F0 and F1 groups was not significantly different from conventional As3mt-KO mice ($p = 0.3335$, Mantel–Cox test). These results establish in vivo evidence that a human gut microbiome provides protection against arsenic-induced mortality.

**Donor-dependent variation in arsenic protection.** Human gut microbiomes vary greatly in species composition between people. We therefore tested whether host protection varies as a function

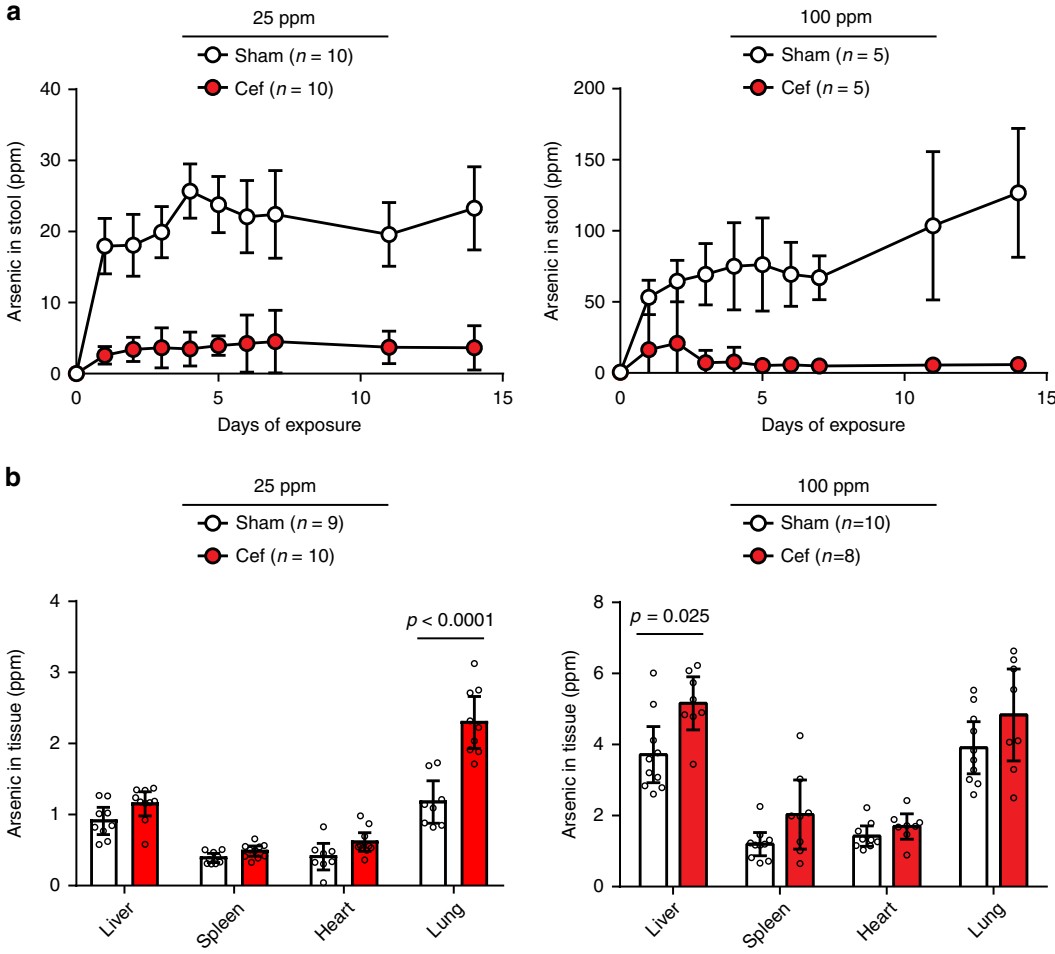

**Fig. 1** Arsenic levels in feces and host tissues of wild-type C57BL/6 mice. Cefoperazone (Cef) and Sham treatments were initiated 48 h prior to inorganic sodium arsenate ($iAs^V$) exposure in drinking water at either 25 ppm (left) or 100 ppm (right). For 25 ppm exposures, a total of 20 mice were used in two replicate experiments ($n = 5$ mice per treatment group; tissue from one mouse in Sham group was lost during processing). For 100 ppm exposures (right), a total of 10 mice were used to examine fecal excretion ($n = 5$, Sham; $n = 5$, Cef) and an additional (replicate) experiment was added for tissue bioaccumulation (an additional 8 mice; $n = 5$, Sham; $n = 3$, Cef). Group-wise means are shown in all graphs (scatter plots and histograms) and error bars represent 95% confidence interval of the mean. **a** Significantly lower levels of arsenic were excreted in feces of Cef-treated mice beginning 24 h after exposure at both doses ($p < 0.0001$, two-way ANOVA after log transformation). **b** Significantly higher levels of arsenic accumulated in organs of Cef-treated mice after 14 days of exposure (25 ppm, $p < 0.0001$; 100 ppm, $p = 0.0003$, two-way ANOVA after square root transformation) with the greatest differences in lung (25 ppm) and liver (100 ppm). $p$-Values in **b** were adjusted for multiple comparisons (Sidak's multiple comparisons test)

of inter-individual microbiome composition using five groups of $iAs^V$-exposed GF As3mt-KO mice that received stool transplants from different healthy adult donors. GF As3mt-KO mice were humanized and exposed to 100 ppm $iAs^V$ as described above (i.e. F0 mice). All humanized microbiome groups survived significantly longer than the GF As3mt-KO group ($p ≤ 0.0014$, Mantel–Cox test, Supplementary Table 2). However, significantly different survival was observed between humanized groups of mice ($p < 0.0001$, Mantel–Cox test) with median survival times ranging from 17 to 36 days (Fig. 3b; Supplementary Table 3). Independent (replicate) experiments were performed using two donor groups; one that provided better protection than the other. Although median survival varied somewhat (Supplementary Fig. 2), the overall effect was reproducible (i.e., median survival was greater in one group in both replicates; aggregate data for replicates were used for all analyses). Thus, human stool transplantation revealed inter-individual differences in protection from arsenic toxicity.

To identify patterns of bacterial diversity underlying survival, we conducted 16S rRNA gene sequencing on stool samples from humanized mice at the outset of arsenic exposure (day 0) and again at day 7 prior to any mortality. Rarefaction analysis indicated that reasonable sequencing coverage was obtained in all samples (Supplementary Fig. 3). Humanized mouse microbiomes were significantly more similar within donor groups (i.e., mice sharing the same donor) than between donor groups ($p = 0.001$, ANOSIM; Fig. 4a). Despite these donor-specific patterns, significant changes in microbiome structure (i.e., presence–absence and relative abundance of taxa) was observed in all groups between days 0 and 7 of arsenic exposure ($p ≤ 0.013$, ANOSIM; Supplementary Table 4). These results support previous studies in mice[5,6,18] that arsenic perturbs the gut microbiome; however, the extent of perturbation between groups was significantly different ($p < 0.0001$, Kruskal–Wallis test), suggesting that perturbation is dependent on the starting microbiome (Fig. 4b). The magnitude of perturbation between days 0 and 7 in each donor group can be considered a measure of community stability (i.e., stability—little change). In survival models (Cox Proportional Hazards; CPH) that account for differences within donor groups, an increase in alpha diversity (both Shannon and inverse Simpson diversity) was significantly correlated with longer survival (Shannon, $p = 0.0447$; inverse Simpson, $p = 0.0155$). Likewise, stability significantly correlated

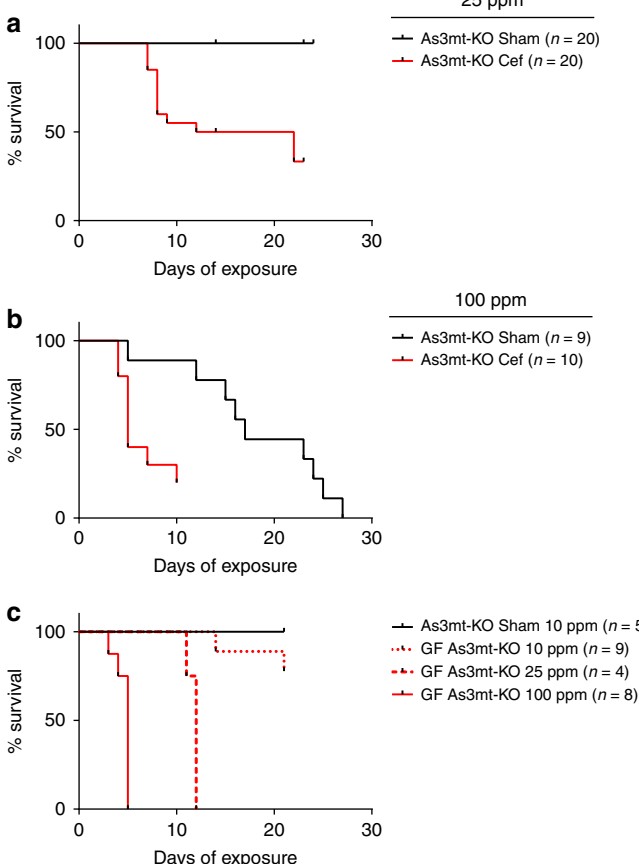

**Fig. 2** Survival of iAs$^V$-exposed Cef-treated, Sham-treated, and GF As3mt-KO mice. Cef and Sham treatments were initiated 48 h prior to iAs$^V$ exposure in drinking water at either 25 ppm (**a**) or 100 ppm (**b**). **a** For 25 ppm exposure, a total of 40 mice were used in four replicate experiments (five mice per treatment group). **b** For 100 ppm exposure, a total of 19 mice were used in two replicate experiments (four or five mice per treatment group). Survival of Cef-treated mice was significantly lower compared to Sham-treated mice at both 25 ppm ($p < 0.0001$, Mantel–Cox test) and 100 ppm ($p = 0.0032$, Mantel–Cox test) exposures. For GF As3mt-KO mice (**c**), three groups were exposed to increasing levels of iAs$^V$ (i.e., two replicate cages of four or five mice were exposed to 10 and 100 ppm exposures, and one cage of four mice was exposed to 25 ppm) along with a negative control group of conventional As3mt-KO mice (one cage of five mice). Dose-dependent survival was observed among these groups. Although mortality was observed in GF As3mt-KO mice exposed to 10 ppm, survival was not significantly different compared to the conventional, Sham-treated As3mt-KO mice exposed to 10 ppm ($p = 0.2274$, Mantel–Cox test). Survival was significantly lower in GF As3mt-KO mice exposed to both 25 ppm ($p = 0.0050$, Mantel–Cox test) and 100 ppm ($p = 0.0014$, Mantel–Cox test) compared to conventional, Sham-treated As3mt-KO mice. Survival was also significantly lower in GF As3mt-KO mice exposed to 100 ppm compared to 25 ppm ($p = 0.0031$, Mantel–Cox test)

with survival ($p = 0.0325$, Wald test). Also, these covariate models performed significantly better than the reduced model (i.e., donor group only, $p < 0.05$, ANOVA), indicating the importance of such diversity estimates within donor groups. Overall, these results suggest that the ability of the gut microbiome to maintain taxonomic integrity under arsenic stress is important for host survival.

**Individual microbiome members correlate with survival.** Gut bacteria that metabolize arsenic into less toxic arsenicals or

provide some other detoxification properties could potentially be beneficial during exposure. We attempted to identify these bacteria with survival modeling of the 100 most abundant operational taxonomic units (OTUs) observed among all humanized mice. Considering OTU presence/absence at days 0 and 7, and controlling for false discoveries ($q < 0.1$), 48 unique OTUs were identified as being associated with survival either negatively or positively (Fig. 5). Of these 48 taxa, 22 were significantly associated with survival at both days 0 and 7 (shared), 21 were only associated at day 0 (Fig. 5a), and 5 were only associated with day 7 (Fig. 5b). All OTU-survival associations shared on both days were in agreement with respect to coefficient sign (i.e., positive or negative), supporting a consistent direction of their influence. A complementary analysis was performed using log-transformed OTU abundance data for days 0 and 7 with similar results (Supplementary Fig. 4). This analysis identified 53 significant associations between OTU abundance and host survival, with 30 being shared (i.e., significantly on both days). A total of 38 (72%) of the OTUs associated with survival using relative abundance data were also identified in the presence/absence analysis, including 17 of the 22 (77%) shared OTUs. These results suggest that both the presence/absence and relative abundance of specific microbiome members influence survival during arsenic exposure.

***Faecalibacterium prausnitzii* provides protection.** Interpreting the above results is somewhat difficult due to co-evolutionary relationships that lead to the co-occurrence of microbiome members within and between donor groups. In other words, such analyses only establish correlative relationships, and remain insufficient to resolve causality between specific taxa and in vivo function. To determine causality for at least one microbiome member and to establish whether our model of arsenic toxicity is sensitive enough to detect the influence of a single bacterium, gnotobiotic experiments with *F. prausnitzii* in As3mt-KO mice were conducted. The rationale for selecting this particular taxon is as follows: *Faecalibacterium* is one of the most abundant and commonly identified taxa in the gut microbiome of healthy humans, of which *F. prausnitzii* is the only named species. *F. prausnitzii* contributes to butyrate and other short-chain fatty acid production in the gut[19], and a reduction in *F. prausnitzii* has been associated with a wide range of gastrointestinal diseases, including inflammatory bowel disease[20], colorectal cancer[21,22], and *Clostridium difficile* infection[23]. Analysis of *F. prausnitzii* genomes (NCBI) revealed the presence of an iAs$^{III}$ S-adenosylmethyltransferase gene, *arsM*, in at least some genomes, which encodes a well-known methyltransferase involved in microbial arsenic detoxification[24]. A *Faecalibacterium* OTU was detected in the microbiome of 36 out of the 44 humanized mice; it was present at both days (0 and 7) across all donor groups; and was abundant, ranging from 0.01% to 5.3% of normalized reads. Survival modeling (both presence/abundance and abundance analyses) identified this OTU to be significantly associated with survival at both days 0 and 7 (Fig. 5, Supplementary Fig. 4). Finally, *F. prausnitzii* is already marketed as a human probiotic, thus providing a clear connection to human health. The combination of results from previous studies, results from humanized mice, the potential for production of an arsenic-active enzyme(s), and current marketability led us to hypothesize that *F. prausnitzii* provides protection against arsenic toxicity.

To test the above hypothesis, gnotobiotic mouse experiments were conducted using *F. prausnitzii* strain, A2-165, that was originally isolated from a healthy human. Miquel et al. previously found that A2-165 did not readily colonize GF mice when administered alone, but colonized at a high level (~10$^8$ CFU per gram of stool) when administered in combination with *Escherichia coli*[25]. This was also observed in the current study; i.e., A2-165 could not be detected in stool of GF mice 48 h post-gavage, whereas it became abundant in stool of *E. coli* mono-colonized

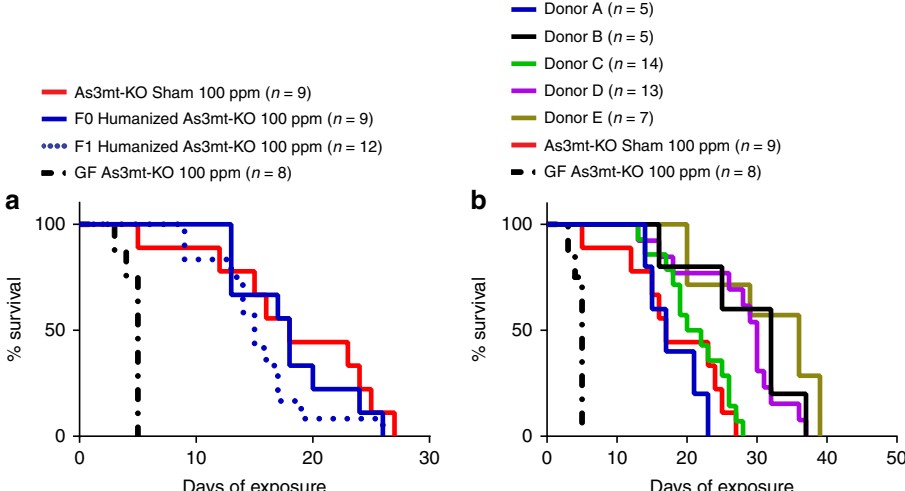

**Fig. 3** Survival of iAs$^V$-exposed humanized As3mt-KO mice. Groups of As3mt-KO mice (i.e., two or three replicate cages containing between two and five mice each) received a fecal transplant from a healthy human donor (humanized) either 10 days prior to iAs$^V$ exposure (F0) or at birth from a humanized dam (F1). **a** Survival of F0 and F1 groups was not significantly different compared to conventional, Sham-treated As3mt-KO mice following exposure to 100 ppm iAs$^V$ (F0, $p = 0.6500$; F1, $p = 0.1887$; Mantel–Cox test), but survival in both groups was significantly greater than GF As3mt-KO ($p < 0.0001$, Mantel–Cox test). Five groups of humanized As3mt-KO mice, each representing a unique human microbiome donor, were exposed to 100 ppm iAs$^V$ (two to four replicate cages per group). **b** Median survival in all donor groups was significantly greater than that of GF As3mt-KO mice ($p \leq 0.0014$, pairwise Mantel–Cox), and equivalent to or greater than that of Sham-treated conventional mice. There were also significant differences in survival among different humanized groups ($p = 0.0030$, Mantel–Cox). Pairwise comparisons (Mantel–Cox test) are shown in Supplementary Table 4. Survival of GF and conventional, Sham-treated As3mt-KO from previous figures (Fig. 2c and b, respectively) are shown in both panels for comparison

mice. In a series of experiments, mono-colonization of GF mice with *E. coli* strains K-12 (W3110) or B (BL21) did not provide significant protection during 25 ppm iAs$^V$ exposure (Supplementary Fig. 5). However, *E. coli*–*F. prausnitzii* bi-colonization significantly increased survival compared to both GF ($p = 0.0035$) and *E. coli* mono-associated mice ($p = 0.0024$, Fig. 6). These results support the hypothesis that *F. prausnitzii* is sufficient for at least partial protection against arsenic toxicity, and also suggest that the GF As3mt-KO model of arsenic exposure is highly sensitive to detect these effects. Future studies are now needed to determine whether this effect is due to in vivo production of active arsenic enzymes (i.e., metabolism), production of non-arsenic-related metabolites (e.g., short-chain fatty acids), or some other host–microbiome factor(s).

## Discussion

Mammalian cells and many microbes detoxify arsenic, but the in vivo role of the human microbiome in detoxifying arsenic has not been addressed. Experiments summarized herein show that either microbiome perturbation or absence increases host arsenic bioaccumulation and toxicity. Further, GF As3mt-KO mice were shown to be a useful, hypersensitive arsenicosis model and that human stool transplantation restores protection in these mice. Protection was at least partially due to microbiome composition and stability, providing the first in vivo evidence that the human microbiome protects against arsenic toxicity. Finally, *F. prausnitzii* appears to be a useful correlate of microbiome stability during arsenic exposure in humans, and also provides some protection in the GF As3mt-KO murine model. Although microbes can enzymatically transform arsenical compounds, they can also bioaccumulate this toxin[26–30], which may facilitate excretion and help limit host exposure. Thus, future research is needed to quantify the net influence of these two potentially important mechanisms for arsenic detoxification in the human body. Based on these results, we propose that the microbiome should be a target for arsenicosis prevention and treatment

strategies, and considered a plausible explanatory factor in epidemiologic studies that attempt to account for the often observed variability in disease penetrance among similarly exposed individuals[31]. Given the scale and scope of global arsenicosis, probiotics with active arsenic metabolisms capable of mitigating arsenic toxicity may represent feasible, low-cost therapeutics.

## Methods

**Experimental animals**. Animal experiments were approved by the Montana State University Institutional Animal Care and Use Committee. All mice were bred and maintained at an American Association for the Accreditation of Laboratory Animal Care accredited facility at Montana State University. Conventional mice were housed under specific pathogen-free conditions (including murine norovirus) in individually ventilated cages with sterilized bedding. GF mice were housed in hermetically sealed and Hepa-filter ventilated, vinyl isolators and received autoclaved water and food (LabDiet, St. Louis, MO). All food and water for GF mice were quarantined , monitored, and tested for contamination prior to introduction. No statistical methods were used to select a priori sample sizes, no randomization techniques were used, and no investigator blinding was done.

Only C57BL/6 mice were used, being either wild-type or deficient for the murine *as3mt* gene (As3mt-KO). Breeding pairs of As3mt-KO mice were obtained from Drs. Lora L. Arnold and Samuel M. Cohen (University of Nebraska Medical Center) and Drs. Christelle Douillet and Mirek Styblo (University of North Carolina), and used to establish a breeding colony at Montana State University. These mice were originally derived at the Environmental Protection Agency[15]. GF colonies of both WT and As3mt-KO mice were derived from conventional mice. Briefly, conventional and GF surrogate (Swiss-Webster) females were gestationally synchronized so that litters were born within 12 h of each other. At precisely full term, gravid conventional dams were euthanized, and a sterile hysterectomy was performed. Neonatal pups were quickly removed, revived, and transferred into a GF isolator where they nursed on surrogate GF Swiss-Webster dams. GF status was monitored regularly by attempting aerobic and anaerobic culture techniques on rich medium (Mueller–Hinton broth and agar plates) and by periodic PCRs (Supplementary Table 6) targeting the bacterial 16S rRNA encoding gene with DNA extracted from stool (see below) serving as a template.

**Human samples**. Human fecal samples collected and cryopreserved as part of an unrelated study were used for humanization experiments to test for inter-individual variation in the protective effect of human microbiome communities in GF As3mt-KO mice. A total of nine volunteers were originally recruited according to a research protocol approved by the Institutional Review Board of Montana State University in Bozeman, MT, USA. All participants were enrolled with

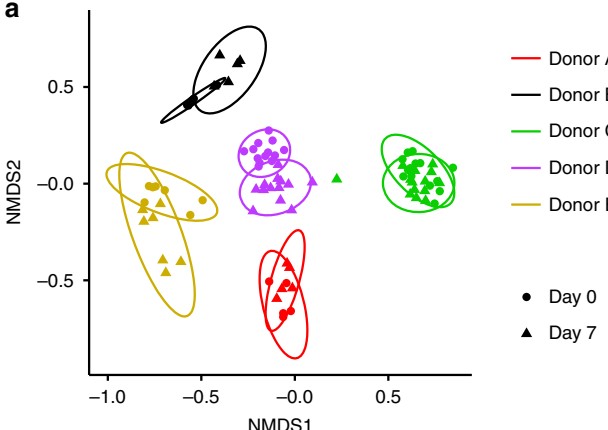

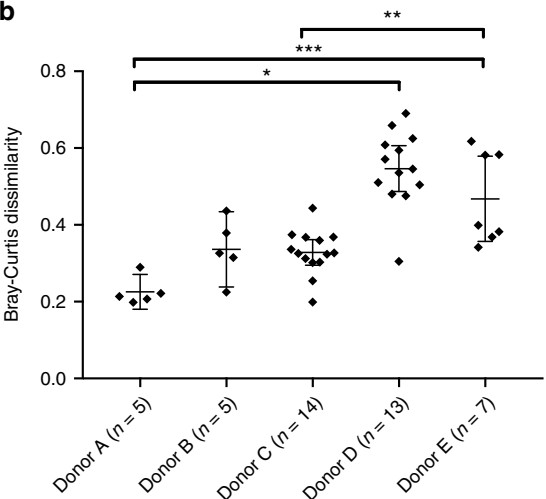

**Fig. 4** Differences in microbiome stability during arsenic exposure. **a** Non-metric multidimensional scaling of Bray–Curtis dissimilarity illustrates both within and between group as well as within and between exposure time changes in microbiome diversity. Each point represents a single humanized mouse microbiome according to donor (color) and time-point (symbol). Ellipses represent 95% confidence. Microbiomes were more similar within donor groups than between ($p = 0.001$, ANOSIM), and in all donor groups, day 7 microbiomes were significantly different from baseline (day 0) microbiomes ($p \leq 0.009$, ANOSIM; Supplementary Table 5). **b** Stability of humanized mouse microbiomes during iAs$^V$ exposure (mean and 95% confidence interval) was significantly different between groups as shown (*$p < 0.01$; **$p < 0.005$, ***$p < 0.0001$, Kruskal–Wallis test)

informed consent, and the only inclusion criterion was that the individual was ≥21 years of age. No information regarding health or medical status of volunteers was collected. Fecal samples were self-collected using a disposable Commode Specimen Collection System (Fisherbrand), transferred into 50 ml screw-cap tubes (~5 g) using a sterile tongue depressor, refrigerated at 4 °C for up to 2 h before processing. Under anaerobic conditions (anaerobic chamber, Coy Laboratories), ~5 g of fresh donor stool sample was mixed with ~30 ml sterile, pre-reduced PBS, mixed with 15% glycerol (final concentration), aliquoted into sterile 1.5 ml cryogenic gasket-cap vials (Neptune Scientific), and stored at −80 °C prior to use in humanization experiments. Archived samples were used from 6 donors (5 male, 1 female) ranging in age from 24 to 40 at the time of sample collection.

**Chemical reagents**. Cefoperazone (cefoperazone sodium salt) was purchased from Chem-Impex International Inc. (Wood Dale, IL) and stored with desiccant at 4 °C. iAs$^V$ was ACS grade (≥98% pure) sodium arsenate dibasic heptahydrate (Na$_2$HAsO$_4$·7H$_2$0, Sigma-Aldrich, St. Louis, MO). All chemicals were used as received with no additional purity analysis.

**Arsenic exposures**. Except where indicated (see Supplementary Table 5), age-matched mice (7–13 weeks old) of both sexes were used. In rare cases, the decision to use mice >13 weeks of age was based solely on mouse availability. To disrupt the microbiome of mice, cefoperazone (Cef, 0.5 mg per ml) was added to drinking water 48 h prior to and throughout arsenic exposure. Sham-treated groups received drinking water containing an identical concentration of Cef that was denatured by boiling for 5 min and cooled to room temperature before use. Denaturation was confirmed using a bioassay with a sensitive *E. coli* strain (MG1655) in LB broth. Fresh water was made for both Cef- and Sham-treated groups every 3–4 days. All arsenic exposures were as inorganic sodium arsenate (iAs$^V$ at either 25 or 100 ppm). Fecal pellets were collected daily and frozen at −80 °C for quantification of total arsenic. After 14 days of iAs$^V$ exposure, surviving mice were humanly euthanized by isoflurane overdose and tissue was collected for arsenic quantification. GF WT and As3mt-KO mice were exposed to 10, 25, or 100 ppm iAs$^V$ in drinking water. Most exposures were continued until complete mortality or when mice appeared severely moribund at which point they were humanely euthanized.

**Fecal transplantation**. GF As3mt-KO recipient mice received a fecal transplant from human donors prior to 100 ppm iAs$^V$ exposure via drinking water. We chose this model because it was found that mortality occurred gradually after 14 days in these mice, thus allowing for statistical comparisons of median survival. Frozen human donor stool samples (see above) were thawed inside an anaerobic chamber, suspended in sterile, pre-reduced PBS at approximately a 1:1 weight to volume ratio, and 100 µL of this slurry was used to inoculate GF mice via oral gavage. Humanized mice were allowed to equilibrate for 10–14 days prior to iAs$^V$ exposure. In addition to humanizing naive adult GF mice (F0), we created a cohort of As3mt-KO mice that were humanized from birth (F1) by humanizing GF As3mt-KO dams prior to breeding. All experiments with humanized mice were conducted inside sterile, vinyl isolators.

**Arsenic quantification**. Arsenic concentrations in biological samples were determined using an Agilent 7500 ICP-MS. Mouse tissue and fecal pellet samples were weighed and digested in a 70% solution of trace-metal grade nitric acid (VWR International, Radner, PA) using heat and pressure (115 °C, 29.7 PSI for 30 min). Digested samples were diluted in ultra-pure water to achieve a final nitric acid concentration of 5%. Final dilutions were centrifuged at ~2000 rcf (Beckman GS-6R centrifuge) for 10 min to remove particulates, and the supernatant was collected for analysis. Samples were injected via a constant-flow-rate peristaltic auto-sampler and the results quantified against an external standard curve using the Agilent's ChemStation software package.

**16S rRNA gene sequencing**. Groups of up to 3–5 mice were co-housed in the same cage during exposure studies. Cages were assigned to treatment groups, while attempting to control for sex as a biological variable. Fecal pellets were collected by holding mice above the cage and collecting pellets directly into sterile Eppendorf tubes (i.e., mice reproducibly defecate when held). Bulk DNA was extracted from fecal pellets using the DNeasy® PowerSoil® Kit (Qiagen, Hilden, Germany) and stored at −20 °C. DNA was shipped overnight on dry ice to the University of Michigan, Center for Microbial Systems, for Illumina MiSeq amplicon sequencing of the 16S V4 variable region. After DNA quantification, V4 amplicon libraries were generated with dual-index barcoded primers (Supplementary Table 6), followed by library purification, pooling, and MiSeq paired-end sequencing using 2 × 250 base-pair chemistry.

Raw sequencing reads were processed and curated using the mothur (v.1.39.5) software package[32], following the mothur MiSeq standard operating procedure[33] accessed on May 17, 2017 (http://www.mothur.org/wiki/MiSeq_SOP). Paired-end reads were assembled into contigs, and screened for length and quality (minimum 253 bp and no ambiguous base calls). The remaining contigs were aligned to coordinates 6427 through 25319 of the Silva ribosomal RNA gene reference database, release 128[34,35]. Potentially chimeric sequences were identified and removed using the Uchime (v. 4.2.40) algorithm via mothur[36]. Taxonomic classifications were assigned using a Bayesian classifier at the Ribosomal Database Project[37] (v. 14) implemented in mothur, using training set version 16[38]. Reads classifying as non-target organisms/organelles (mitochondria, chloroplast, Eukaryota, and sequences unclassified at the domain level) were removed. OTUs were assigned in mothur using the VSEARCH (v. 2.3.4) distance-based greedy clustering algorithm at the 97% sequence similarity threshold[39], and an OTU-based data matrix was built. Rare OTUs (represented by less than 100 total reads after the identification and removal of chimeras) were removed to minimize the influence of spurious OTUs. Finally, representative OTU sequences were classified to the genus level using the Bayesian classifier at the Ribosomal Database Project[37].

**Gnotobiotic mouse experiments**. *F. prausnitzii*, strain A2-165, was obtained from the German Collection of Microorganisms and Cell Cultures (Leibniz Institute DSMZ) and cultured in YCFAG broth[40]. *E. coli*, strains W3110 and BL21, were obtained from Dr. Harry Mobley (University of Michigan) and New England Biolabs, Inc., respectively, and cultured in LB broth. GF As3mt-KO recipient mice received either 10$^8$ colony forming units (CFUs) of *E. coli* alone or in combination

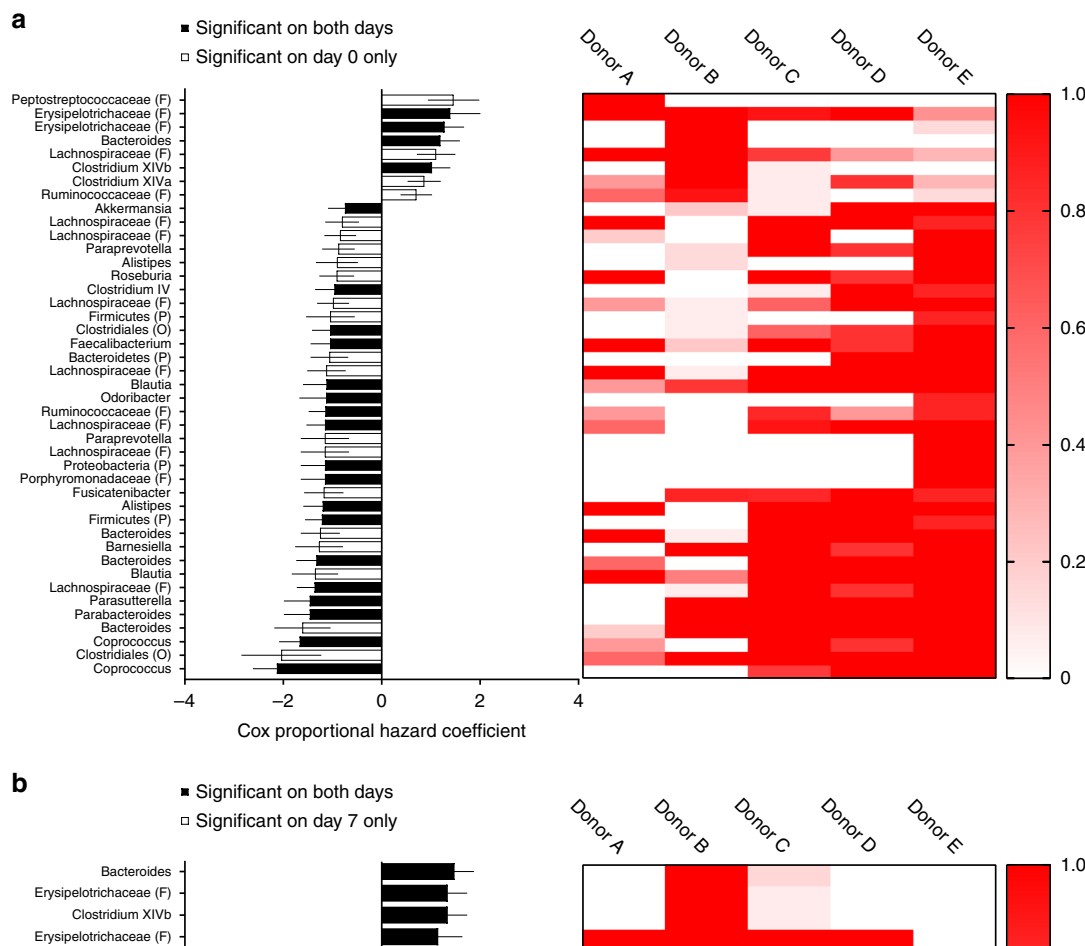

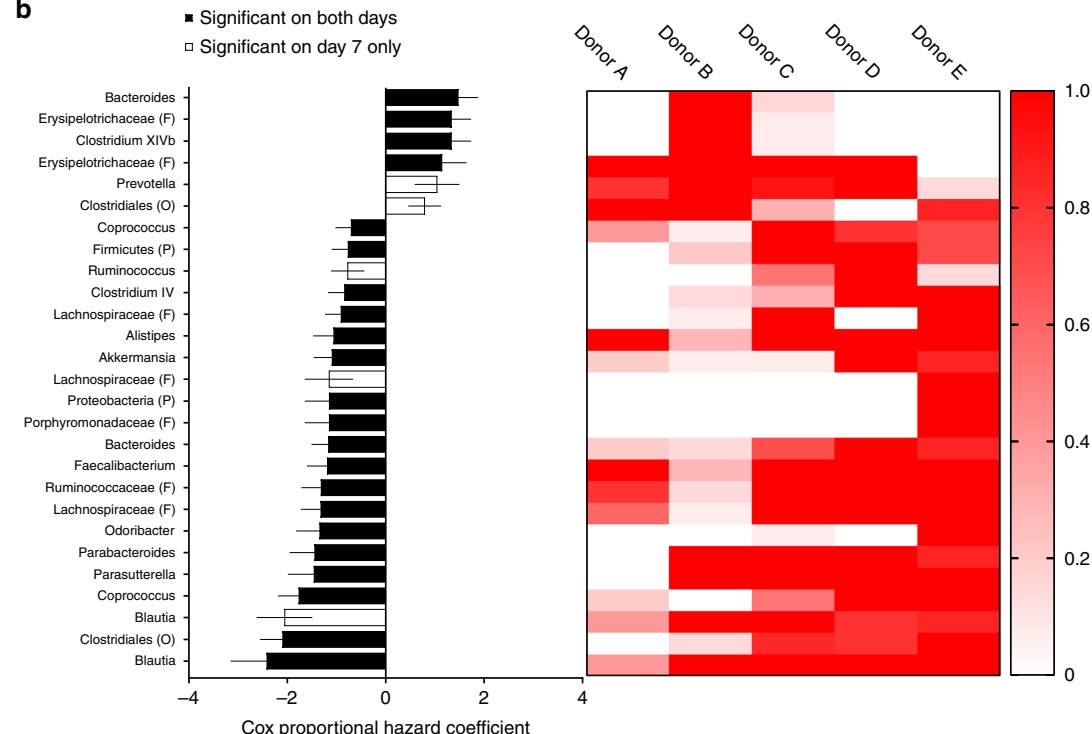

**Fig. 5** Specific microbiome OTUs are risk factors for survival. The presence/absence of the 100 most abundant OTUs in the dataset were fit to univariate Cox proportional hazards (CPH) models. **a**, **b** Controlling for false discovery rate (<0.1), significant beta coefficients (with standard error) for microbiome members at day 0 (**a**, left panel) and day 7 (**b**, left panel) are shown along with the group-wise prevalence of each OTU displayed as heat maps (**a**, **b**, right panels). Donor groups in heat maps (A–E) are ordered by median survival during arsenic exposure (shortest survival on left, longest survival on right). Significant OTUs found on both sampling days (0 and 7) are shown with black bars and significant OTUs found to be unique on one day or the other are shown in open bars. CPH models correlate explanatory variables with a relative hazard function, therefore, a negative beta coefficient indicates a protective association, while positive coefficients are associated with increased hazard. OTUs were classified to the lowest taxonomic level at a 95% confidence. Taxonomic levels above genera are indicated in parentheses (F family, O order, P phylum)

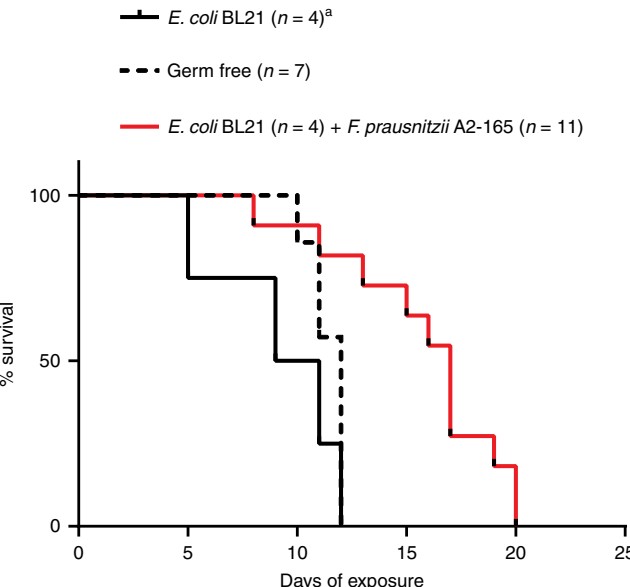

**Fig. 6** *F. prausnitzii* protects against arsenic toxicity. Germ-free, *E. coli* mono-colonized (one cage of four mice), and *E. coli* + *F. prausnitzii* bi-colonized (three replicate cages of 3 or 5 mice each) As3mt-KO mice were exposed to 25 ppm iAs$^V$. Survival was not significantly different between GF and *E. coli* mono-colonized groups ($p = 0.1759$, Mantel–Cox test), whereas mice bi-colonized with *E. coli* + *F. prausnitzii* survived significantly longer than both GF ($p = 0.0035$, Mantel–Cox test) and *E. coli* mono-colonized mice (0.0024, Mantel–Cox test)

with $10^8$ CFUs of *F. prausnitzii* (bi-colonization) via oral gavage. For bi-colonization, *E. coli* was administered to mice 24 h prior to *F. prausnitzii*. A second gavage of *F. prausnitzii* was administered 24 h after the first. To increase *F. prausnitzii* establishment, sodium bicarbonate (0.1 M) was administered just prior to *F. prausnitzii* gavage. Colonization by both *E. coli* and *F. prausnitzii* was confirmed by direct plating from stool onto either MacConkey (to inhibit *F. prausnitzii*) or YCFAG agar containing nalidixic acid (15 µg per ml; to inhibit *E. coli*). Colony PCR was performed on *F. prausnitzii* isolates using species-specific primers[41] to confirm taxonomic identification (Supplementary Table 6). In rare cases, where no *F. prausnitzii* was detected in individual mouse pellets collected 48 h after the initial gavage, a third gavage of sodium bicarbonate and *F. prausnitzii* culture were performed. Colonization was confirmed for at least 72 h prior to arsenic exposure.

**Statistical analyses**. Arsenic levels in feces and organs of mice were either log or square-root transformed, respectively, prior to statistical testing and passed the Shapiro–Wilk normality test (GraphPad Prism 7.03 for Windows, GraphPad Software, La Jolla, CA, USA; www.graphpad.com). A two-way repeated measures ANOVA was used to test for group-wise differences in fecal excretion throughout arsenic exposure. An ordinary two-way ANOVA was used to test for group-wise differences in organ arsenic content. Mouse survival between groups during arsenic exposure was tested using the Mantel–Cox test (GraphPad Prism). Rarefaction, alpha diversity estimates (Shannon and Inverse Simpson), comparisons of beta diversity (Bray–Curtis dissimilarity), and ordinations (e.g., NMDS plots) of 16S rRNA gene sequences were conducted in R[42] (v. 3.3.3) using R packages *vegan*[43] (v. 2.4-5) and *LabDSV*[44] (v. 1.8-0) as implemented in RStudio[45] (v 1.1.383). Statistical analyses and graphs were generated using R package *ggplot2*[46] (v. 2.2.1) and GraphPad Prism (GraphPad Software, La Jolla, CA, USA; www.graphpad.com).

For the purposes of this analysis, Bray–Curtis distance between individual mouse microbiome communities sampled on either day 0 or day 7 of arsenic exposure was used as a proxy for microbiome stability or community change (with more stable communities having smaller BC distance). Differences in microbiome stability between groups of mice were tested using the Kruskal–Wallis test (GraphPad Prism). Analysis of similarities (ANOSIM), as implemented in the R package *vegan*[43], was used to test for differences in microbiome communities between donor groups and different time points within donor groups. Statistical models testing covariate associations with survival were performed using CPH modeling, implemented using the *survival* package[47] (v. 2.38) in R. This technique models predictor variables against a function of proportional hazard instead of a survival function. Therefore, a negative association (beta coefficient < 0) indicates

increased protection or survival in the subjects (lower hazard = greater expected survival). The significance of model coefficients was tested using the Wald test, while over model significance was tested using the Likelihood Ratio (LR) test. All *p*-values represent 2-tailed tests unless otherwise noted in text or figure legends.

## Data availability

All 16S rRNA sequencing reads were deposited in the National Center for Biotechnology Information (NCBI) BioProject database with accession code PRJNA486373. The authors declare that all other data supporting the findings of the study are available in this article and its Supplementary Information files, or from the corresponding author upon request.

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

## Acknowledgements

The authors are grateful to L.L. Arnold and S.M. Cohen at the University of Nebraska Medical Center and C. Douillet and M. Styblo (University of North Carolina) for the gift of the As3mt-KO mice. The authors would also like to thank J. Borgogna at Montana State University for assisting with survival modeling. This research was supported by the work performed by The University of Michigan Microbial Systems Molecular Biology Laboratory. Research reported in this publication was supported by the National Institute of Environmental Health Sciences of the National Institutes of Health (NIH) under Award Numbers R21ES026411 and F31ES026884, the National Institutes of General Medical Sciences and the National Cancer Institute (NIH) under Award Number R01CA215784, and the National Institute of Food and Agriculture, U.S. Department of Agriculture, Hatch project 1009600.

## Author contributions

S.T.W. and T.R.M. conceived the project. S.T.W. directed the research. M.C., M.M., and N.V.P. collected data. M.C., T.R.M., and S.T.W. wrote the paper with comments from all authors. The content is solely the responsibility of the authors and does not necessarily represent the official views of the National Institutes of Health.

## Additional information

**Competing interests:** The authors declare no competing interests.



