## [Peer Review File · Nature Communications]

Reviewers' comments:

Reviewer #1 (Remarks to the Author):

This is a very interesting study to investigate the role of gut microbiome in mediating arsenic toxicity. The authors used a number of animal models and fecal transplant experiments to demonstrate that human microbiome reduces arsenic toxicity. The reviewer has the following comments for the authors to consider to improve the manuscript for publication.

1. What is the justification to use 25 and 100 ppm of arsenic? They are way higher than environmentally or human relevant doses, which are at ppb levels. The authors mentioned a few points in the manuscript, but the major body of arsenic research in animal studies does not commonly use these high doses, except some cancer bioassays.
2. Some details of experiments are not provided. For example, how the mice were housed? single? paired? How were fecal pellets collected for 16s rRNA gene sequencing?, etc.
3. Some data may be difficult to be interpreted. In Figure 1, in 25 ppm samples, lung has about 2 fold higher arsenic in the tissue. But in 100 ppm samples, a marginal increase was found in liver but not lung. Any explanation? In addition, it is conceivable that reduced bacteria amount arising from antibiotics would decrease arsenic load in feces, as bacteria could transport arsenic into cells. Could the authors calculate what is the total amount of arsenic in feces vs total tissues? Thus, the quantitative contributions of microbiome and host metabolism would be generated.
4. The confounding factor was not addressed: CEF was used throughout arsenic exposure; However, sham-treated mice received denatured CEF. Did the authors have the control group with active CEF only? Is there any possibility that active CEF and arsenic interact and affect the readouts? How to prove or disprove this?
5. The authors identified that human gut bacteria, *Faecalibacterium*, may contribute to arsenic detoxification. Unfortunately, no functional data were provided to support such an important link. Could the authors colonize this bacteria in the mice to examine its role in arsenic toxicity?

Reviewer #2 (Remarks to the Author):

In this manuscript, the authors describe the use of a series of murine models to evaluate the effects of the gut microbiome on arsenic toxicity. The authors demonstrate attenuated toxicity in the presence of an intact microbiome, both in wild-type animals and those specifically deficient in arsenate methyltransferase (a primary, host-associated arsenic detoxification pathway). The authors also demonstrate variable arsenic toxicity attenuation in humanized mice as a function of donor and the relative abundance of *Faecalibacterium*.

This is an elegantly-designed, clearly written study that demonstrates a clear link between the gut microbiome and arsenic toxicity. Although generally well-written, it would benefit from the consideration of the following:

- 1) I would encourage the authors to consider an alternative title. Although the current title, "The human gut microbiome protects against arsenic toxicity" does represent the outcomes of the study, my concern is that it has the potential to be over-interpreted or misconstrued and lead to the misconception that the human gut microbiome is wholly protective from arsenic toxicity (e.g., in humans as well as in the mouse model(s) described), when epidemiological data (and some of the human donors involved in this study) demonstrate that arsenic exposure and toxicity remain important global health issues.

2)The placement of references in the abstract seems a bit strange to me. That said, I have not confirmed whether this is in line with the journal's formatting guide.

3)Figure 3b: Given that the authors provide per-donor results later in the manuscript, it would be helpful to see the survivor curves presented in such a way that the survivorship related to each donor can be identified. Having all donor curves in blue does not allow for this.

4)Figure 1: The caption refers to panels a, b, c, and d, but only a and b appear in the figure.

5)With respect to the statistical analysis of the 16S rRNA gene data, were the read counts transformed, subsampled, or normalized in any other way prior to analysis? Please specify. Likewise, were assumptions of normality met with the use of parametric tests (e.g., paired t-test in Supplemental Figure 3)?

Response to Reviewers' comments

Below we provide responses to the Reviewer comments and concerns, point-for-point. In addition, from the perspective of having mentally rested the manuscript for roughly three months during the review process, we now find a few instances where minor corrections are required and we include those as well.

Reviewer #1

Comment

“This is a very interesting study to investigate the role of gut microbiome in mediating arsenic toxicity. The authors used a number of animal models and fecal transplant experiments to demonstrate that human microbiome reduces arsenic toxicity. The reviewer has the following comments for the authors to consider to improve the manuscript for publication.

1. What is the justification to use 25 and 100 ppm of arsenic? They are way higher than environmentally or human relevant doses, which are at ppb levels. The authors mentioned a few points in the manuscript, but the major body of arsenic research in animal studies does not commonly use these high doses, except some cancer bioassays.”

Response

Selection of the dosing levels in our study (10, 25, and 100 ppm) was based on three criteria. First, for reasons highlighted below, acute arsenic toxicity studies in mice typically use high doses of inorganic arsenate (iAs^V) or arsenite (iAs^{III}), even though the majority of humans are naturally and chronically exposed to much lower levels. The range of iAs^V dosing in our study was intentionally consistent with other studies using C57BL/6 WT and As3mt-KO mice (see Table 1 below). We also draw attention to the fact that a similar range of dosing in mice (1-50 ppm) was used to develop a *human* pharmacokinetic (PK) and pharmacodynamic modeling framework at the 2007 Annual Meeting of the Society of Toxicology¹. Consequently, in terms of dosing *per se*, the levels we used are superimposable to those historically used.

Second, if the focus of a study is on toxicity, the level of exposure should be above the lowest-observed-adverse-effect-level (LOAEL) and somewhat lower than the immediate lethal dose (LD₅₀). Unfortunately, there is considerable variability in LOAEL and LD₅₀ estimates for humans, but according to the ASTDR and EPA's toxicological profile on arsenic², the LOAEL for acute, oral exposure (Appendix A in profile) is approximately 0.05 mg As/kg/day. Assuming (as recommended by ASTDR²) a 55 kg person drinking 4.5 L of water per day and a 0.002 mg As/kg/day daily food intake, this level of exposure equates to 611 ppb arsenic in drinking water. Also according to the ASTDR, oral arsenic exposures in drinking water >60,000 ppb can result in death. Thus, exposures between 0.6 - 60 ppm should be appropriate for studying acute toxicity.

Third, an allometric conversion must be done to normalize dosages between humans and animals. For example, the FDA suggests³ that a human equivalent dose (HED) in drug exposure studies is:

$$HED \left(\frac{mg}{kg} \right) = Animal \ dose \left(\frac{mg}{kg} \right) \text{ multiplied by } \frac{Animal \ Km}{Human \ Km}$$

where Km values are the ratio of body weight to surface area (mouse Km = 3, human Km = 37)⁴. Using this equation, 10, 25, and 100 ppm exposures in mice correspond to HEDs of 811, 2,027, and 8,108 ppb in humans. All three of these exposures are well within the ASTDR toxicity range described above. In addition, the first level (811 ppb) is actually lower than that of drinking water recently reported for an arsenic rich part of Chile⁵. The middle and upper exposure levels represent doses where we expected to see increasing (dose-dependent) levels of toxicity.

In summary, the combination of previously published studies, the ASTDR-defined human toxicity range, and allometric conversions between mouse and human support the levels of exposure used in our study. These details have been summarized in the revised manuscript (see lines 45-52), and more details, including the table below, have been added as Extended Data Table 1.

Table 1. Typical range of doses considered in murine arsenic exposure studies.

Arsenical	Mouse (C57BL/6 = WT)	Dose (ppm)^a	Reference
iAs ^V	A/J	1, 10, 100	Cui et al ⁶
iAs ^{III}	WT	0.01, 0.25	Dheer et al ⁷
iAs ^{III}	WT	18.75, 37.5, 62.5	Garcia-Montalvo et al ⁸
iAs ^{III}	WT	10	Lu et al ⁹
iAs ^{III}	WT, IL10-KO	10	Lu et al ¹⁰
iAs ^{III}	WT	10	Lu et al ¹¹
iAs ^{III}	CD1	6, 12, 24	Tokar et al ¹²
iAs ^{III}	CD1	0.05, 0.5, 5	Waalkes et al ¹³
iAs ^{III} , iAs ^V	WT, As3mt-KO	25, 100	Dodmane et al ¹⁴
iAs ^V	WT, As3mt-KO	3.125	Naranmandura et al ¹⁵
iAs ^V	WT, As3mt-KO	3.125	Drobna et al ¹⁶
iAs ^V	WT, As3mt-KO	3.125	Hughes et al ¹⁷
iAs ^{III}	WT, As3mt-KO	25	Arnold et al ¹⁸
iAs ^{III}	WT, As3mt-KO	1, 10, 25, 50	Yokohira et al ¹⁹
iAs ^{III}	WT, As3mt-KO	50, 100, 150	Yokohira et al ²⁰

^aExposures reported by Garcia-Montalvo et al, Naranmandura et al, Drobna et al, and Hughes et al were converted from mg As/kg body weight/day to ppm in water based on a 20 gram mouse drinking 3.2 mL per day⁸.

Comment

2. Some details of experiments are not provided. For example, how the mice were housed? single? paired? How was fecal pellets collected for 16s rRNA gene sequencing?, etc.

Response

More details have been added to the methods section (see 16S rRNA encoding gene sequencing section). To directly address the question here, groups of up to 3-5 mice were co-housed in the same cage during exposure studies. Cages were assigned based on treatment. Fecal pellets were collected by holding mice above the cage and collecting the pellets directly into sterile Eppendorf tubes (i.e. mice reproducibly defecate when held).

Comment

3. Some data may be difficult to be interpreted. In Figure 1, in 25 ppm samples, lung has about 2 fold higher arsenic in the the tissue. But in 100 ppm samples, a marginal increase was found in liver but not lung. Any explanation? In addition, it is conceivable that reduced bacteria amount arising from antibiotics would decrease arsenic load in feces, as bacteria could transport arsenic into cells. Could the authors calculate what is the total amount of arsenic in feces vs total tissues? Thus, the quantitative contributions of microibome and host metabolism would be generated.

Response

We agree that bioaccumulation was not consistently increased in the same organ(s) of cefoperazone-treated mice in experiments at both 25 and 100 ppm. We have not yet followed up on this interesting observation, but it seems reasonable that our sample size ($n \leq 10$) was simply too low to achieve statistical significance in this two-way experimental design (four tissues by two treatments). In support of this, the mean level of arsenic in all organs of cefoperazone-treated mice was more than sham-treated mice, regardless of exposure level (significant overall effect of treatment; all red bars greater than white bars in Figure 1). This suggests that accumulation differences were real, even though only a single organ in either experiment achieved significance. Additional studies are necessary to test the hypothesis that the some organs accumulate more arsenic than others. Since the treatment effect (cefoperazone vs. sham) was highly significant ($p \leq 0.003$) in both experiments, we feel that our conclusion (cefoperazone-treated mice accumulated more arsenic in their organs) was well supported. We can see how this may have been confusing, and have revised the text regarding Figure 1 to help clarify this point. See lines 55-61 for the new text.

Regarding microbiome biomass reduction accounting for the observed reduction in fecal arsenic loads, Reviewer #1 brings up an important point, to which we are in complete agreement. However, for reasons of brevity for this journal (i.e. word count) we did not elaborate on this issue in our original submission. There is actually substantive evidence in the literature that microbes indeed bioaccumulate arsenic, and as such will almost certainly influence total arsenic in the stool and consequently host exposure. We see this as easily varying as a function of gut microbiome composition, although there is not yet enough data on this topic to allow for anything except conjecture at this stage. Nevertheless, it deserves at least a brief commentary and we have added text on lines 175-178, where we address this possibility and provide a few additional citations in support.

Comment

4. The confounding factor was not addressed: CEF was used throughout arsenic exposure; However, sham-treated mice received denatured CEF. Did the authors have the control group with active CEF only? Is there any possibility that active CEF and arsenic interact and affect the readouts? How to prove or disapprove this?

Response

These are excellent points. In our original study, we did not include a Cef-only control group, as no observable effects (toxicity or mortality) were reported in previous studies of Cef-treated mice^{21,22}. In addition, results in figure 2 show that Cef-treatment did not add to the toxicity of arsenic. For example, the median survival of Cef-treated, conventional As3mt-KO mice at both 25 (panel a) and 100 (panel b) ppm were very similar to the median survival of germ free animals

exposed to the same arsenic levels (panel c). This is difficult to explain if CEF was toxic by itself or if it significantly altered arsenic toxicity.

However, the point that Cef may synergistically influence arsenic toxicity is important to address. To do so, we conducted additional experiments. First, we pretreated conventional As3mt-KO mice with Cef for varying lengths of time prior to 100 ppm iAs^V exposure (Day -7 and -2) or on the same day (Day 0) as iAs^V exposure (Extended Data Section Figure 1, panel a). Survival in mice pretreated with Cef at Day -2 was the shortest (median = 5 days) and were completely consistent with previous results (Figure 2, panel b). In contrast, survival of mice that were pre-treated with Cef for 7 days (median = 15 days) or that received Cef on the same day that iAs^V exposure began (median = 15) was not significantly different from mice that did not receive Cef (median = 17 days). These results suggest that the antibiotic's influence on toxicity was greatest when given 2 days prior to iAs^V exposure, which coincides with microbiome disruption reported in previous studies with this drug^{21,22}. The lack of a significant difference in survival when Cef was administered on Day -7 was presumably due to the ability of the microbiome to rebound or normalize from the initial dysbiosis. Similarly, the lack of difference in survival when Cef was administered on Day 0 was presumably due the pharmacokinetics/pharmacodynamics of the antibiotic, meaning that the microbiome was likely to provide some initial protection before the effects of the drug were realized. More experiments will be needed to address these two new hypotheses. However, these results are difficult to explain if synergism between Cef and iAs^V has a significant effect on toxicity.

Second, we treated a group of germ free As3mt-KO mice with Cef and iAs^V at 25 ppm. As shown originally in Figure 2 panel c, treatment of GF As3mt-KO mice with 25 ppm iAs^V results in a median group survival of 12 days. Thus, we reasoned that this timeframe would allow us to observe whether our 2-day Cef pretreatment would decrease survival upon iAs^V exposure (Extended Data Section Figure 1 panel b). In fact, the median survival of this group of mice was 18 days, which is slightly longer than GF As3mt-KO mice. The difference in median survival (18 vs. 12 days) could be due to a variety of factors (e.g. water consumption), but speculating on the mechanism at this point would be complete conjecture. We also note that our original comparisons used denatured Cef (Sham) control, which may control for at least some of this difference. Regardless, all available results provide evidence that active Cef does not increase the toxicity of arsenic in these murine models.

Comment

5. The authors identified that human gut bacteria, *Faecalibacterium*, may contribute to arsenic detoxification. Unfortunately, no functional data were provided to support such an important link. Could the authors colonize this bacteria in the mice to examine its role in arsenic toxicity?

Response

We agree that this is another important aspect to address. The only known species of *Faecalibacterium* found in humans is *F. prausnitzii*, and a rather straight-forward experimental approach would have been to compare survival between GF and *F. prausnitzii* mono-colonized As3mt-KO mice during iAs^V exposure. However, a recent study with a representative *F. prausnitzii* strain, A2-165, found that it did not readily colonize germ free (GF) mice²³, but required pre-colonization with *E. coli* as a mutualistic partner. Thus, we attempted to compare survival between groups of GF, *E. coli* mono-colonized, and *E. coli* + *F. prausnitzii* bi-colonized As3mt-KO mice. Fortunately for us, we had already conducted numerous iAs^V exposure experiments with GF and

E. coli mono-colonized As3mt-KO mice as part of this and other ongoing projects. These data contribute to this manuscript, but also help reduce yet additional animal experimentation. Our *E. coli* mono-colonization experiments considered survival in mice individually colonized with two different *E. coli* strains (W3110, a K12 derivative; and BL21, an *E. coli* B derivative), and neither provided significant protection compared to GF mice when exposed to 25 ppm iAs^v (Extended Data Section Figure 2, panel b). The only experimental variation in these studies was that both *E. coli* strains had carried empty cloning vectors (plasmids) that were maintained in mice by adding antibiotics to drinking water. However, since neither strain provided protection to mice compared to their GF, non-antibiotic treated counterparts, we feel it is appropriate to compare survival between these groups and *E. coli* + *F. prausnitzii* bi-colonized mice. Based on two experiments (total n=11), we found that *F. prausnitzii* provided significant protection to both GF and *E. coli* mono-colonized mice (Extended Data Section Figure 2, panel b). Therefore, these new results support the hypothesis initially raised during our human microbiome analysis that *Faecalibacterium* provides protection during arsenic exposure. A new panel (panel e) has been added to Figure 4 showing the survival of GF, *E. coli* BL21 mono-colonized, and *E. coli* BL21 + *F. prausnitzii* A2-165 bi-colonized mice, and text has been added in lines 153-166.

Reviewer #2:

Comment

In this manuscript, the authors describe the use of a series of murine models to evaluate the effects of the gut microbiome on arsenic toxicity. The authors demonstrate attenuated toxicity in the presence of an intact microbiome, both in wild-type animals and those specifically deficient in arsenate methyltransferase (a primary, host-associated arsenic detoxification pathway). The authors also demonstrate variable arsenic toxicity attenuation in humanized mice as a function of donor and the relative abundance of *Faecalibacterium*.

This is an elegantly-designed, clearly written study that demonstrates a clear link between the gut microbiome and arsenic toxicity. Although generally well-written, it would benefit from the consideration of the following:

1) I would encourage the authors to consider an alternative title. Although the current title, “The human gut microbiome protects against arsenic toxicity” does represent the outcomes of the study, my concern is that it has the potential to be over-interpreted or misconstrued and lead to the misconception that the human gut microbiome is wholly protective from arsenic toxicity (e.g., in humans as well as in the mouse model(s) described), when epidemiological data (and some of the human donors involved in this study) demonstrate that arsenic exposure and toxicity remain important global health issues.

Response

We appreciate this comment and have edited the title to hopefully minimize any negative impact on current global health efforts.

Comment

2)The placement of references in the abstract seems a bit strange to me. That said, I have not confirmed whether this is in line with the journal's formatting guide.

Response

A referenced abstract is consistent (recommended) by Nature's formatting guide.

Comment

3)Figure 3b: Given that the authors provide per-donor results later in the manuscript, it would be helpful to see the survivor curves presented in such a way that the survivorship related to each donor can be identified. Having all donor curves in blue does not allow for this.

Response

We revised figure 3 panel b so that each donor can be clearly distinguished.

Comment

4)Figure 1: The caption refers to panels a, b, c, and d, but only a and b appear in the figure.

Response

This is an unfortunate and regrettable typographical error on our part, and the legend has been revised accordingly. We thank Reviewer #2 for bringing this issue to our attention.

Comment

5)With respect to the statistical analysis of the 16S rRNA gene data, were the read counts transformed, subsampled, or normalized in any other way prior to analysis? Please specify. Likewise, were assumptions of normality met with the use of parametric tests (e.g., paired t-test in Supplemental Figure 3)?

Response

More details have been added to the Methods section regarding the 16S dataset and processing. Prior to analysis, the numbers of OTUs and phylotypes in each library were normalized by evenly subsampling each community with mothur to a depth of 33,166 (i.e. the abundance of total reads in the library with fewest number of reads). This procedure was done according to the mothur SOP for MiSeq analysis. Normality was met for all data used in parametric testing with only a few exceptions in Extended Data Figure 4, top panel (*Faecalibacterium*). The p-values for these particular comparisons have been removed and noted.

References

- 1 Kenyon, E. M. *et al.* How can biologically-based modeling of arsenic kinetics and dynamics inform the risk assessment process? - A workshop review. *Toxicology and applied pharmacology* **232**, 359-368, doi:10.1016/j.taap.2008.06.023 (2008).
- 2 ATSDR. (ed Health and Human Services) (Atlanta, GA, 2007).
- 3 Center for Drug Evaluation and Research, C. f. B. E. a. R. (ed US Food and Drug Administration) (Rockville, MD, USA, 2002).
- 4 Freireich, E. J., Gehan, E. A., Rall, D. P., Schmidt, L. H. & Skipper, H. E. Quantitative comparison of toxicity of anticancer agents in mouse, rat, hamster, dog, monkey, and man. *Cancer chemotherapy reports* **50**, 219-244 (1966).
- 5 Apata, M., Arriaza, B., Llop, E. & Moraga, M. Human adaptation to arsenic in Andean populations of the Atacama Desert. *American journal of physical anthropology* **163**, 192-199, doi:10.1002/ajpa.23193 (2017).
- 6 Cui, X., Wakai, T., Shirai, Y., Hatakeyama, K. & Hirano, S. Chronic oral exposure to inorganic arsenate interferes with methylation status of p16INK4a and RASSF1A and induces lung cancer in A/J mice. *Toxicological sciences : an official journal of the Society of Toxicology* **91**, 372-381, doi:10.1093/toxsci/kfj159 (2006).
- 7 Dheer, R. *et al.* Arsenic induces structural and compositional colonic microbiome change and promotes host nitrogen and amino acid metabolism. *Toxicology and applied pharmacology* **289**, 397-408, doi:10.1016/j.taap.2015.10.020 (2015).
- 8 Garcia-Montalvo, E. A., Valenzuela, O. L., Sanchez-Pena, L. C., Albores, A. & Del Razo, L. M. Dose-dependent urinary phenotype of inorganic arsenic methylation in mice with a focus on trivalent methylated metabolites. *Toxicology mechanisms and methods* **21**, 649-655, doi:10.3109/15376516.2011.603765 (2011).
- 9 Lu, K. *et al.* Gut microbiome perturbations induced by bacterial infection affect arsenic biotransformation. *Chemical research in toxicology*, doi:10.1021/tx4002868 (2013).
- 10 Lu, K. *et al.* Gut microbiome phenotypes driven by host genetics affect arsenic metabolism. *Chemical research in toxicology* **27**, 172-174, doi:10.1021/tx400454z (2014).
- 11 Lu, K. *et al.* Arsenic exposure perturbs the gut microbiome and its metabolic profile in mice: an integrated metagenomics and metabolomics analysis. *Environmental health perspectives* **122**, 284-291, doi:10.1289/ehp.1307429 (2014).
- 12 Tokar, E. J., Diwan, B. A., Ward, J. M., Delker, D. A. & Waalkes, M. P. Carcinogenic effects of "whole-life" exposure to inorganic arsenic in CD1 mice. *Toxicological sciences : an official journal of the Society of Toxicology* **119**, 73-83, doi:10.1093/toxsci/kfq315 (2011).
- 13 Waalkes, M. P., Qu, W., Tokar, E. J., Kissling, G. E. & Dixon, D. Lung tumors in mice induced by "whole-life" inorganic arsenic exposure at human-relevant doses. *Archives of toxicology* **88**, 1619-1629, doi:10.1007/s00204-014-1305-8 (2014).
- 14 Dodmane, P. R. *et al.* Characterization of intracellular inclusions in the urothelium of mice exposed to inorganic arsenic. *Toxicological sciences : an official journal of the Society of Toxicology* **137**, 36-46, doi:10.1093/toxsci/kft227 (2014).
- 15 Naranmandura, H., Rehman, K., Le, X. C. & Thomas, D. J. Formation of methylated oxyarsenicals and thioarsenicals in wild-type and arsenic (+3 oxidation state) methyltransferase knockout mice exposed to arsenate. *Analytical and bioanalytical chemistry* **405**, 1885-1891, doi:10.1007/s00216-012-6207-0 (2013).
- 16 Drobna, Z. *et al.* Disruption of the arsenic (+3 oxidation state) methyltransferase gene in the mouse alters the phenotype for methylation of arsenic and affects distribution and retention of

- orally administered arsenate. *Chemical research in toxicology* **22**, 1713-1720, doi:10.1021/tx900179r (2009).
- 17 Hughes, M. F. *et al.* Arsenic (+3 oxidation state) methyltransferase genotype affects steady-state distribution and clearance of arsenic in arsenate-treated mice. *Toxicology and applied pharmacology* **249**, 217-223, doi:10.1016/j.taap.2010.09.017 (2010).
- 18 Arnold, L. L. *et al.* Time Course of Urothelial Changes in Rats and Mice Orally Administered Arsenite. *Toxicologic pathology*, doi:10.1177/0192623313489778 (2013).
- 19 Yokohira, M. *et al.* Effect of sodium arsenite dose administered in the drinking water on the urinary bladder epithelium of female arsenic (+3 oxidation state) methyltransferase knockout mice. *Toxicological sciences : an official journal of the Society of Toxicology* **121**, 257-266, doi:10.1093/toxsci/kfr051 (2011).
- 20 Yokohira, M. *et al.* Severe systemic toxicity and urinary bladder cytotoxicity and regenerative hyperplasia induced by arsenite in arsenic (+3 oxidation state) methyltransferase knockout mice. A preliminary report. *Toxicology and applied pharmacology* **246**, 1-7, doi:10.1016/j.taap.2010.04.013 (2010).
- 21 Antonopoulos, D. A. *et al.* Reproducible community dynamics of the gastrointestinal microbiota following antibiotic perturbation. *Infection and immunity* **77**, 2367-2375, doi:10.1128/IAI.01520-08 (2009).
- 22 Reeves, A. E. *et al.* The interplay between microbiome dynamics and pathogen dynamics in a murine model of *Clostridium difficile* Infection. *Gut microbes* **2**, 145-158 (2011).
- 23 Miquel, S. *et al.* Identification of metabolic signatures linked to anti-inflammatory effects of *Faecalibacterium prausnitzii*. *mBio* **6**, doi:10.1128/mBio.00300-15 (2015).

REVIEWERS' COMMENTS:

Reviewer #1 (Remarks to the Author):

The authors have done a great job to address most of comments raised by the reviewer. The reviewer strongly supports the publication of this work in the Journal. One minor comment, it seems that the authors did not point out whether male or female mice were used in the study, or both sexes?

Reviewer #2 (Remarks to the Author):

In this revised manuscript, the authors have addressed several of the concerns raised during the initial review process. The additional data presented is helpful, as is the inclusion of additional details regarding methods. That said, the manuscript would still benefit from the consideration of the following:

- 1) Although the author addressed previous queries regarding 16S sequence processing and analysis methods, the text of the methods section could better reflect this information. For example, although the authors cite the mothur MiSeq SOP, it might not be clear to many readers that this approach explicitly includes the use of even-depth subsampling prior to statistical analysis. Likewise, the inclusion of a section specifically describing the statistical test used (i.e., not just including them in figure legends) would be helpful to interested readers.
- 2) In figure 1 (all panels), please specify the unit(s) of variance captured by the error bars.
- 3) In the caption for figure 3, panel a is not explicitly identified.
- 4) In the caption for figure 4, it is stated that, "Stability was negatively correlated with survival (c)," and "The stability of Faecalibacterium was negatively correlated with survival (d)." These interpretations appear to be the opposite of the trends displayed by the data, and those which are described in the main body of the text. In panel C, the most stable communities are those which share the least Bray-Curtis dissimilarity between the two time points tested (i.e., the values to the far right of the figure). These data suggest a positive correlation with stability, rather than a negative one. Similarly, in panel D, the communities with the greatest stability of Faecalibacterium and survival rates are those appearing on the far right-hand side of the figure (i.e., least change in Faecalibacterium abundance), again indicating that stability is positively correlated with survival.

Point-by-point response to reviewers' comments

Reviewer #1 (Remarks to the Author):

The authors have done a great job to address most of comments raised by the reviewer. The reviewer strongly supports the publication of this work in the Journal. One minor comment, it seems that the authors did not point out whether male or female mice were used in the study, or both sexes?

Response: The sex of mice used in all experiments has been clarified in the methods section (Arsenic exposures) and identified per experiment in the supplementary information (Supplementary Table 2).

Reviewer #2 (Remarks to the Author):

In this revised manuscript, the authors have addressed several of the concerns raised during the initial review process. The additional data presented is helpful, as is the inclusion of additional details regarding methods. That said, the manuscript would still benefit from the consideration of the following:

1) Although the author addressed previous queries regarding 16S sequence processing and analysis methods, the text of the methods section could better reflect this information. For example, although the authors cite the mothur MiSeq SOP, it might not be clear to many readers that this approach explicitly includes the use of even-depth subsampling prior to statistical analysis. Likewise, the inclusion of a section specifically describing the statistical test used (i.e., not just including them in figure legends) would be helpful to interested readers.

Response: We agree that additional information/clarification on the statistical analyses was needed and we added a section to the methods specifically addressing this point.

2) In figure 1 (all panels), please specify the unit(s) of variance captured by the error bars.

Response: This detail has been added to all figures where error bars are shown.

3) In the caption for figure 3, panel a is not explicitly identified.

Response: All panels for figures have been explicitly identified in the revised manuscript.

4) In the caption for figure 4, it is stated that, "Stability was negatively correlated with survival (c)," and "The stability of Faecalibacterium was negatively correlated with survival (d)." These interpretations appear to be the opposite of the trends displayed by the data, and those which are described in the main body of the text. In panel C, the most stable communities are those which share the least Bray-Curtis dissimilarity between the two time points tested (i.e., the values to the far right of the figure). These data suggest a positive correlation with stability, rather than a negative one. Similarly, in panel D, the communities with the greatest stability of Faecalibacterium and survival rates are those appearing

on the far right-hand side of the figure (i.e., least change in Faecalibacterium abundance), again indicating that stability is positively correlated with survival.

Response: We realize that the use of “positive” and “negative” may be confusing. Statisticians typically use these terms to describe the tandem movement of variables, where “positive” refers to variables that increase or decrease together. This was not the case in our study because Bray-Curtis is a dissimilarity metric (1=not stable; 0=stable) and the fold decrease in Faecalibacterium was plotted instead of fold increase. To avoid this confusion, we simply state that significant correlations were observed and then explain what the correlations mean (e.g. “...mice with more stable microbiomes were better protected.”). We hope that this is more straight-forward for readers.

Editorial Note: After discovering a regulatory issue with the human stool samples used in the study, the authors replaced all data generated on these samples with similar samples for which they had full approval to work with, granted by their local Institutional Regulatory Review Board.

REVIEWERS' COMMENTS:

Reviewer #1 (Remarks to the Author):

In this manuscript, the authors provide evidence that an intact gut microbiome confers resistance to toxic arsenic exposures and identify *Faecalibacterium* a key contributor to this effect. The results of this study are quite exciting and will be of interest to the greater community.

In this revised submission, the authors present a clear rationale for the work that they did, and the level of methodological detail provided would allow for replication of most aspects of this work. In order to move the bar to the point of replication of all aspects, I would encourage the authors to:

- 1. Provide additional detail regarding the recruitment of the human donors used in this study (i.e., What, if any, exclusion criteria were applied? Within what age range did the donors fall? How many males vs. females were recruited?) Please also include the number of human subjects recruited in the methods section. Although one surmises that n=5 subjects from the figures and tables, it would be preferable to see this presented concretely in the methods section.**
- 2. Specify what read chemistry was used to generate the 16S reads. One presumes that it was 2x250 bp chemistry, but this is not explicitly stated.**
- 3. Mo Bio was acquired by Qiagen a few years ago. Please provide current manufacturer information for the DNA extraction kits that were used.**
- 4. In describing the evaluation of microbiome stability (e.g., lines 374-375), it would be helpful to remind the reader that this refers to changes or differences in Bray-Curtis distances. "Microbiome stability", per se, is not a term that many in the microbiome field (or otherwise) might automatically recognize and understand.**

Reviewer #2 (Remarks to the Author):

The revised manuscript essentially reported the same finding as the previous version, although different donors were used. New experiments can be viewed as validation. This actually strengthened the study. The reviewer supports the publication of this revised manuscript in the Journal.

Response to reviewer comments

REVIEWERS' COMMENTS:

Reviewer #1 (Remarks to the Author):

In this manuscript, the authors provide evidence that an intact gut microbiome confers resistance to toxic arsenic exposures and identify *Faecalibacterium* a key contributor to this effect. The results of this study are quite exciting and will be of interest to the greater community.

In this revised submission, the authors present a clear rationale for the work that they did, and the level of methodological detail provided would allow for replication of most aspects of this work. In order to move the bar to the point of replication of all aspects, I would encourage the authors to:

1. Provide additional detail regarding the recruitment of the human donors used in this study (i.e., What, if any, exclusion criteria were applied? Within what age range did the donors fall? How many males vs. females were recruited?) Please also include the number of human subjects recruited in the methods section. Although one surmises that $n=5$ subjects from the figures and tables, it would be preferable to see this presented concretely in the methods section.
2. Specify what read chemistry was used to generate the 16S reads. One presumes that it was 2x250 bp chemistry, but this is not explicitly stated.
3. Mo Bio was acquired by Qiagen a few years ago. Please provide current manufacturer information for the DNA extraction kits that were used.
4. In describing the evaluation of microbiome stability (e.g., lines 374-375), it would be helpful to remind the reader that this refers to changes or differences in Bray-Curtis distances. "Microbiome stability", per se, is not a term that many in the microbiome field (or otherwise) might automatically recognize and understand.

Responses:

1. More detail has been added to the Methods section of the manuscript regarding human donor recruitment. Additionally, the methods used in sample collection and preparation have been updated to allow for possible replication by others in the field.
2. The chemistry used to generate 16s sequence reads has been specified explicitly in the methods section of the manuscript.
3. Manufacturer information for DNA extraction kits used has been updated.

4. A statement clarifying our use of Bray-Curtis distance as a proxy for microbiome stability has been added to the manuscript.

Reviewer #2 (Remarks to the Author):

The revised manuscript essentially reported the same finding as the previous version, although different donors were used. New experiments can be viewed as validation. This actually strengthened the study. The reviewer supports the publication of this revised manuscript in the Journal.

No response required